# Ubiquitination of GRK2 Is Required for the β-Arrestin-Biased Signaling Pathway of Dopamine D2 Receptors to Activate ERK Kinases

**DOI:** 10.3390/ijms241210031

**Published:** 2023-06-12

**Authors:** Haiping Liu, Haixiang Ma, Xingyue Zeng, Chengyan Wu, Srijan Acharya, Sarabjeet Kour Sudan, Xiaohan Zhang

**Affiliations:** 1School of Pharmaceutical Sciences, Guizhou University, Guiyang 550025, China; hpliu0508@126.com (H.L.); mt863256957@163.com (H.M.); zengxingyue0906@163.com (X.Z.); cywu0624@163.com (C.W.); 2Mitchell Cancer Institute, School of Medicine, University of South Alabama, Mobile, AL 36604, USA; sacharya@southalabama.edu (S.A.); sksudan@southalbama.edu (S.K.S.)

**Keywords:** GRK2 ubiquitination, D2R-biased signaling, Src, ERK signaling

## Abstract

A class-A GPCR dopamine D2 receptor (D2R) plays a critical role in the proper functioning of neuronal circuits through the downstream activation of both G-protein- and β-arrestin-dependent signaling pathways. Understanding the signaling pathways downstream of D2R is critical for developing effective therapies with which to treat dopamine (DA)-related disorders such as Parkinson’s disease and schizophrenia. Extensive studies have focused on the regulation of D2R-mediated extracellular-signal-regulated kinase (ERK) 1/2 signaling; however, the manner in which ERKs are activated upon the stimulation of a specific signaling pathway of D2R remains unclear. The present study conducted a variety of experimental techniques, including loss-of-function experiments, site-directed mutagenesis, and the determination of protein interactions, in order to investigate the mechanisms underlying β-arrestin-biased signaling-pathway-mediated ERK activation. We found that the stimulation of the D2R β-arrestin signaling pathway caused Mdm2, an E3 ubiquitin ligase, to move from the nucleus to the cytoplasm and interact with tyrosine phosphorylated G-protein-coupled receptor kinase 2 (GRK2), which was facilitated by Src, a non-receptor tyrosine kinase. This interaction led to the ubiquitination of GRK2, which then moved to the plasma membrane and interacted with activated D2R, followed by the phosphorylation of D2R as well as the mediation of ERK activation. In conclusion, Mdm2-mediated GRK2 ubiquitination, which is selectively triggered by the stimulation of the D2R β-arrestin signaling pathway, is necessary for GRK2 membrane translocation and its interaction with D2R, which in turn mediates downstream ERK signaling. This study is primarily novel and provides essential information with which to better understand the detailed mechanisms of D2R-dependent signaling.

## 1. Introduction

G-protein-coupled receptors (GPCRs) are proteins with a seven-transmembrane domain that are capable of transmitting intracellular signals. They are involved in many physiological processes and are currently one of the most prominent pharmacological targets [1]. It has become evident that a variety of GPCRs preferentially activate distinct G proteins, resulting in signal amplification through the enzymatic activity of second messengers to trigger diverse responses depending on extracellular stimuli [2]. Another signaling mode is associated with a receptor’s ability to activate signaling pathways independently of G protein activation via the scaffolding of β-arrestin, which is a component of GPCR desensitization and internalization machinery [3]. β-arrestin has been demonstrated to connect activated GPCRs to signaling effectors, such as Src tyrosine kinases [4], components of ERKs as well as c-Jun N-terminal kinase 3 (JNK3) [5], and protein kinase B (AKT) [6], effectively conveying G-protein-independent signaling. Since β-arrestin binding can decouple G proteins from an activated receptor, G-protein-dependent as well as β-arrestin-dependent signaling can be distinguished. Additionally, the two signaling pathways, which have been shown to be spatially and temporally different events, can be distinguished pharmacologically [7]. The DA family of GPCRs includes five subtypes, D1–D5, which have partially overlapping functions and pharmacology. D1 and D5 receptors couple to the stimulatory G protein Gs, leading to increased cyclic AMP (cAMP) accumulation, while D2–D4 receptors mainly couple to Gi/o proteins, inhibiting cAMP accumulation [8,9]. In our study, we employed D2R as a model with which to investigate the molecular mechanisms underlying D2R-bias-mediated ERK activation. We chose D2R because it plays a critical role in the development of psychosis, as many antipsychotics act via modulating D2R. Furthermore, D2R is widely expressed in various tissues, including blood vessels as well as the adrenal gland, heart, and kidney, in addition to neurons in the cerebral cortex, olfactory tubercle, and hippocampus [10].

D2R-mediated ERK1/2 signaling, which serves as a critical regulator of numerous fundamental cellular processes, such as differentiation, cell proliferation, and survival, involves complex mechanisms [11]. For example, Phosphatidylinositol 3-kinase (PI3-K), receptor tyrosine kinase (RTK) transactivation, Src kinase, G protein βγ subunits (Gβγ), and protein kinase C (PKC) have been demonstrated to be involved in the D2R-mediated ERK activation that is pertussis toxin (PTX)-sensitive [12,13,14]. PTX catalyzes the ADP ribosylation of the Gi/o α subunit at a cysteine residue four amino acids from the C-terminal end of the protein, resulting in the receptors being uncoupled from the Gi/o proteins. In addition, the effects of β-arrestin and receptor endocytosis on the D2R-mediated activation of ERKs have also been reported, despite the existence of contradictory conclusions [13,15]. Extensive studies have been conducted on the molecular mechanisms underlying D2R-mediated ERK activation; however, it still unclear how ERKs are activated when only one D2R-biased signaling pathway is available.

GRK2 is a ubiquitous member of the GRK family that plays a crucial role in the regulation of GPCRs [16]. In this process, GRK2 is localized in the cytoplasm of unstressed cells, but it translocates to the plasma membrane in response to the agonist stimulation of GPCRs via the direct interaction of its C-terminal plekstrin homology (PH) domain with Gβγ, and then phosphorylates the GPCRs subsequent to recruiting β-arrestin [17]. In fact, the functional role of GRK2 is not limited to facilitating the binding of β-arrestin to activated GPCRs; GRK2 also phosphorylates an increasing number of nonreceptor substrates and interacts with multiple signal-transduction-related proteins [18], and it was reported to play unexpected cellular roles in the modulation of ERK activation [19], the regulation of cell cycle progression [20], or the suppression of transforming growth factor β (TGF-β)-mediated cell growth arrest as well as apoptosis [21]. Demonstrating that GRK2 may have “effector” functions beyond receptor desensitization and endocytosis, GRK2-mediated receptor phosphorylation has acted as a critical step in triggering receptor desensitization, which refers to the attenuation of receptor responsiveness after prolonged or intermittent exposure to agonists, occurring within seconds to minutes, as well as endocytosis, which involves the formation as well as inward cytosolic movement of vesicles originating from the plasma membrane. As a result, activated receptors are internalized within these vesicles and undergo different sorting pathways, which can lead to their degradation or recycling back to the plasma membrane [22].

Alterations in GRK2 protein level and/or activity may have substantial effects on cellular signaling and physiological functions due to the complexity of the GRK2 interactome and its indispensable roles in numerous signaling pathways. Multiple human pathologies have been shown to be related to abnormally high GRK2 protein levels; this kinase is upregulated in portal hypertension, myocardial infarction, insulin resistance, heart failure, and Alzheimer’s disease [16,23,24,25,26]. In contrast, it is downregulated in rheumatoid arthritis [27]. A major mechanism responsible for modulating GRK2 levels is its degradation via the proteasome pathway. Proteasome degradation depends on the coordinated actions of E1 ubiquitin-activating enzyme, E2 ubiquitin conjugating enzymes, and E3 ubiquitin ligases. The E3 ubiquitin ligases, which interact either directly or indirectly with their substrates via adapter molecules, determine the specificity of target protein selection. Mdm2 functions as an E3 ubiquitin ligase that regulates cell growth as well as apoptosis and plays a crucial role in GRK2 ubiquitination as well as degradation [28]. Moreover, the activation of GPCRs has been shown to increase GRK2 ubiquitination and turnover followed by degradation in a proteasome-dependent manner [29] via a mechanism requiring GRK2 phosphorylation through Src or mitogen-activated protein kinase (MAPK) kinase in a β-arrestin-dependent manner [30,31].

In this study, we detected the molecular mechanisms underlying the D2R β-arrestin-dependent pathway that activated ERKs, focusing on the nuclear export of Mdm2, which is responsible for the ubiquitination of GRK2. 

## 2. Results

### 2.1. GRK2-Mediated Receptor Phosphorylation Directs D2R β-Arrestin Signaling-Pathway-Mediated ERK Activation

To evaluate the functional roles of D2R-biased signaling pathway, we generated a G-protein-dependent D2R (^[Gprot]^D2R) and a β-arrestin-dependent D2R (^[βarr]^D2R) through combinatorial amino acid residue substitutions in the second intracellular loop L125N/Y133L or A135R/M140D, respectively [32]. Both of these biased receptors have been previously employed as unique tools for understanding the functions of the biased signaling pathways of D2R [33]. D2R is known to couple with Gi proteins, which inhibit the adenylate cyclase (AC) enzyme and reduce intracellular cAMP levels [34]. In this study, we employed an indirect method with which to detect the suppression of AC and cAMP synthesis induced by D2R. Initially, we exposed the cells to forskolin, a direct activator of the AC enzyme, to induce intracellular cAMP synthesis. In parallel experiments, the cells were exposed to forskolin plus quinpirole (Quin), a D2R agonist. If the cells expressed functional ^[Gprot]^D2R that undergoes a G-protein-dependent pathway, the activation of this receptor by Quin would inhibit forskolin-induced cAMP synthesis. Conversely, the activation of ^[βarr]^D2R, which undergoes a G-protein-independent pathway, would fail to suppress forskolin-induced cAMP synthesis. Next, we used ^[D80A]^D2R as a control, which has been demonstrated to bind ligands and traffic to the plasma membrane, but lacks signaling activity [35]. As shown in Appendix A, ^[Gprot]^D2R retained complete efficacy and potency at cAMP inhibition, whereas ^[βarr]^D2R and ^[D80A]^D2R were severely deficient compared to ^[WT]^D2R. Additionally, the agonist-induced interaction of β-arrestin2 with ^[βarr]^D2R, but not with ^[Gprot]^D2R, increased (Appendix A). These data suggest that ^[Gprot]^D2R or ^[βarr]^D2R can effectively activate the G-protein- or β-arrestin-biased signaling pathway, respectively. 

Since the duration of ERK activation is an important factor in determining its biological responses, we investigated the temporal patterns of ERK activation mediated by the D2R G-protein- or β-arrestin-dependent pathway. Our findings showed that the stimulation of ^[Gprot]^D2R resulted in rapid and transient ERK activation, which decreased within 2 min of agonist treatment. In contrast, the activation of ^[βarr]^D2R led to the relatively delayed but persistent activation of ERKs, which peaked at 10 min and sustained for much longer periods (Appendix A). In the following experiments, we used DA treatment for 2 min or 10 min to detect ERK activation mediated by the D2R G-protein- or β-arrestin-dependent pathway, respectively.

A growing body of evidence demonstrates that GRK2 plays a key role in the control of GPCR-mediated ERK activation [36,37]. We employed GRK2-KD cells to determine the exact roles of GRK2 in DA-mediated ERK activation. As shown in Figure 1A, the knockdown of GRK2 diminished ^[βarr]^D2R-, not ^[Gprot]^D2R-, mediated ERK activation, indicating that GRK2 is responsible for D2R β-arrestin signaling-pathway-mediated ERK activation. It has been previously reported that ERK activation, regardless of GRK2-mediated receptor phosphorylation, one of the best-established cellular events, is the determiner of signaling transduction via GPCRs. For example, GRK2 acts as a RhoA-activated scaffold protein for the ERK MAP kinase cascade or regulation of ERK activity through direct or coordinated interaction with MEK [37,38]. To examine the involvement of GRK2-mediated receptor phosphorylation in D2R β-arrestin pathway-mediated ERK signaling, a specially designed construct, D2R-IC2/3, was then utilized. In D2R-IC2/3, all serine and threonine residues in the second and third intracellular loops of D2R were substituted to alanine and valine residues, respectively, making it beneficial for studying receptor phosphorylation-independent cellular processes [39]. Our data showed that D2R-IC2/3 caused a reduced level of ERK activation upon stimulation via the agonist UNC9994, which effectively activates β-arrestin-biased D2R signaling pathways [40] (Figure 1B). Moreover, GRK2-K220R, a GRK2 mutant devoid of enzymatic activity [41], was employed to determine whether the kinase activity of GRK2 is required for ERK activation. As shown in Figure 1C, ERK activation induced by UNC9994 was rescued only by GRK2-WT, but not by GRK2-K220R, in GRK2-KD cells. These results suggest that the GRK2-mediated phosphorylation of receptors is required for D2R β-arrestin pathway-mediated ERK signaling.

To investigate the functional roles of β-arrestin in D2R G-protein- and β-arrestin-dependent signaling pathways, we employed β-arrestin1/2-KD cells. As expected, ^[βarr]^D2R-, but not ^[Gprot]^D2R-, mediated ERK activation was blocked in β-arrestin1/2-KD cells (Appendix A). There are two isoforms of β-arrestin, β-arrestin1 and β-arrestin2 (also known as arrestin-2 and arrestin-3, respectively), which are ubiquitously expressed and share 78% sequence homology. Previous studies have shown that β-arrestin1 and β-arrestin2 play reciprocal roles in angiotensin II receptor-mediated ERK activation [7]. We then investigated the roles of β-arrestin1 and β-arrestin2 in ^[βarr]^D2R-mediated ERK activation. As shown in Appendix A, the selective knockdown of β-arrestin2 completely abolished ^[βarr]^D2R-mediated ERK activation, whereas the knockdown of β-arrestin1 had no effect, indicating that the β-arrestin2 isoform plays an important role in the D2R-mediated activation of ERKs.

### 2.2. Ubiquitination of GRK2 Is Required for D2R β-Arrestin-Dependent Signaling-Pathway-Mediated ERK Activation

As our results suggest that GRK2 is involved in D2R β-arrestin pathway-mediated ERK activation, we next determined the conditions under which GRK2 escorts β-arrestin pathway-mediated ERK signaling. GRK2 has been shown to undergo Mdm2-mediated ubiquitination [28]. First, Mdm2-KD cells were used to determine whether Mdm2-mediated ubiquitination is involved in ^[βarr]^D2R-mediated ERK activation. As shown in Figure 2A, the knockdown of Mdm2 blocked ^[βarr]^D2R-, not ^[Gprot]^D2R-, mediated ERK activation, suggesting that the Mdm2-mediated ubiquitination of specific cellular components is required for D2R β-arrestin signaling-pathway-mediated ERK activation. When it was tested for GRK2-specific effects, a GRK2 mutant (GRK2-4KR) was generated, in which four lysine residues at positions K19, K21, K30, and K31, which are ubiquitinated by Mdm2, were altered to arginine [28]. Cells producing ^[βarr]^D2R or ^[Gprot]^D2R were transfected with a plasmid encoding either GRK2-WT or GRK2-4KR in GRK2-KD cells. As shown in Figure 2B, GRK2-KD cells producing GRK2-4KR show blocked ^[βarr]^D2R-mediated ERK activation. These results suggest that Mdm2-mediated GRK2 ubiquitination plays an important role in the regulation of D2R β-arrestin pathway-mediated ERK activation.

### 2.3. Activation of the D2R β-Arrestin Signaling Pathway Promotes GRK2 Ubiquitination

Our data showed that GRK2 ubiquitination is involved in D2R β-arrestin pathway-mediated ERK activation; we were interested in determining whether the Mdm2-mediated ubiquitination of GRK2 increased upon stimulation of the D2R β-arrestin signaling pathway. As shown in Figure 3A,B, the activation of ^[βarr]^D2R, not ^[Gprot]^D2R, strongly increased the interaction between GRK2 and Mdm2, with a concomitant increase in GRK2 ubiquitination. Additionally, GRK2 ubiquitination was investigated via employing the biased ligand MLS1547 or UNC9994, which effectively activates G protein- or β-arrestin-biased D2R signaling pathways, respectively [40,42]. As shown in Figure 3C, the stimulation of D2R with UNC9994 led to an increase in the ubiquitination of GRK2. These data suggest that the Mdm2-mediated ubiquitination of GRK2 is controlled by the activation of the D2R β-arrestin signaling pathway. 

In support of the role of ubiquitination in target protein degradation, stimulation with UNC9994 accelerated the decline in cellular GRK2 levels, which was observable 30 min after UNC9994 treatment (Figure 3D); however, this effect was significantly attenuated during the knockdown of Mdm2 (Figure 3E). 

### 2.4. Activation of D2R β-Arrestin Signaling-Pathway-Mediated Mdm2 Nuclear Export

Mdm2 and GRK2 must co-localize in the same intracellular compartment for the Mdm2-mediated ubiquitination of GRK2. Mdm2, an E3 ubiquitin ligase, is located in the nucleus at resting state but translocates to the cytoplasm when certain external stimuli are applied [43]. Previous studies have reported that GRK2 is located in the nucleus of neuronal cells in the striatum [44]; however, it is worth noting that GRK2 contains three modular domains, including the N-terminal RGS homology (RH) domain, the bilobular central kinase domain, and the C-terminal pleckstrin homology (PH) domain [45], but does not have a canonical nuclear localization sequence (NLS) or nuclear export signal (NES) [46]. In the present study, we observed that GRK2 localized in the cytoplasm regardless of pretreatment with leptomycin B (LMB), which has been utilized to prevent NES-mediated nuclear export by interacting with CRM1, a receptor for the NES [47] (Figure 4A), indicating that GRK2 does not shuttle between the cytoplasm and nucleus. Mdm2, which was exclusively localized in the nucleus of resting HEK 293 cells, moved out of the nucleus and was disseminated throughout the nucleus as well as cytoplasm upon D2R β-arrestin pathway activation through UNC9994 (Figure 4B). These results suggest that Mdm2, not GRK2, shuttles between the nucleus and cytoplasm in HEK 293 cells. Thus, it could be speculated that the stimulation of the D2R β-arrestin pathway enhanced Mdm2 nuclear export, resulting in GRK2 ubiquitination in the cytoplasm. To test this hypothesis, Mdm2 was forcibly detained in the nucleus via pretreatment with LMB, and the resulting effects on GRK2 ubiquitination were examined. LMB pretreatment blocked the UNC9994-induced nuclear export of Mdm2 (Appendix A) and GRK2 ubiquitination (Figure 4C), suggesting that Mdm2-mediated GRK2 ubiquitination occurs in the cytoplasm. This conclusion was supported by additional direct biochemical evidence. When cells were treated with UNC9994, GRK2 ubiquitination levels increased in the cytoplasmic, but not the nuclear fraction (Figure 4D). The protein level of GRK2 in the nuclear fraction was much lower than that of the cytoplasmic fraction. This might be due to the fact that GRK2 is a cytoplasmic protein that does not shuttle between the nucleus and cytosol (Figure 4A). The successful separation of nuclear and cytoplasmic fractions has been demonstrated through the absence of β-actin in the isolated nucleus fraction and of laminB1 in the isolated cytosol fractions. 

### 2.5. Src-Mediated GRK2 Tyrosine Phosphorylation Is Required for Mdm2-Mediated GRK2 Ubiquitination

Since non-receptor tyrosine kinase Src has been reported to be necessary for the agonist-induced proteolysis of GRK2 [30], we tested whether the Src-mediated tyrosine phosphorylation of GRK2 is required for its ubiquitination. As shown in Figure 5A,B, pretreatment with PP2 (4-amino-5-(4-chlorophenyl)-7-(t-butyl) pyrazolo[3,4-d]pyrimidine), an ATP-competitive inhibitor of the Src family of protein tyrosine kinases, abolished the ^[βarr]^D2R-mediated interaction of GRK2 with Mdm2 and the ubiquitination of GRK2. Three tyrosine residues (Y13, Y86, and Y92) on GRK2 are phosphorylated by Src [30], all three of which were substituted with phenylalanine residues (3YF) in order to generate non-tyrosine-phosphorylated GRK2. As shown in Figure 5C, GRK2-3YF failed to be ubiquitinated upon the stimulation of ^[βarr]^D2R with DA (Figure 5C). These results suggest that Src-mediated GRK2 tyrosine phosphorylation occurs upstream of ubiquitination modification. Indeed, GRK2-4KR, which failed to be ubiquitinated [28], still interacted with Src in response to DA stimulation (Figure 5D). Overall, these results demonstrate that the Src-dependent tyrosine phosphorylation of GRK2 promotes the interaction of GRK2 with the E3 ubiquitin ligase Mdm2, resulting in the ubiquitination of GRK2.

Next, we asked whether Src-mediated tyrosine phosphorylation influenced GRK2-related D2R β-arrestin pathway-mediated ERK signaling. To answer this question, a plasmid encoding GRK2-3YF was introduced into GRK2-KD cells, and changes in ERK activation were compared with those observed in cells harboring GRK2-WT. GRK2-WT promoted ERK activation, and this activity was lower in GRK2-3YF-producing cells than in a control upon the activation of ^[βarr]^D2R through DA (Figure 5E). 

### 2.6. The Ubiquitination of GRK2 Is Needed for Its Translocation to the Plasma Membrane and Interaction with D2R

Our results showed that the activation of ERKs, following the activation of the D2R β-arrestin signaling pathway, required GRK2-mediated receptor phosphorylation (Figure 1). We hypothesized that agonist-induced interactions between ^[βarr]^D2R and GRK2 would be the critical determinant of this process. As shown in Figure 6A,B, the interaction of GRK2-WT, not GRK2-4KR and GRK2-3YF, with ^[βarr]^D2R increased alongside agonist treatment. The pretreatment of PP2, which abolished GRK2 ubiquitination (Figure 5B), also blocked their interaction (Figure 6C). Additionally, we employed the human neuroblastoma SH-SY5Y cell line as an in vitro model of dopaminergic neurons with which to investigate the interaction between endogenous GRK2 and D2R upon the stimulation of the β-arrestin signaling pathway after pretreatment with the Src kinase inhibitor PP2 (Appendix A). Our data suggest that pretreatment with PP2 blocked the interaction of GRK2 with D2R when stimulated with UNC9994, which is consistent with the conclusions drawn from experiments on HEK 293 cells. 

For interactions with receptors, GRK2 first needs to translocate to the plasma membrane. As shown in Figure 6D (upper panel), GRK2-WT translocated to the cell membrane upon stimulation with UNC9994; however, GRK2-4KR and GRK2-3YF failed to translocate to the plasma membrane (Figure 6D, lower panel). These results, overall, indicate that GRK2 ubiquitination is required for its translocation and interaction with receptors, which subsequently mediate receptor phosphorylation and ERK signaling. 

## 3. Discussion

Although detailed molecular mechanisms involved in D2R-mediated ERK activation have been exclusively reported, the processes involved in D2R-biased signaling-pathway-mediated ERK activation remain unclear. This study showed that, upon the activation of the D2R β-arrestin signaling pathway, Mdm2 translocates out of the nucleus to ubiquitinate GRK2 that has been previously phosphorylated by Src. This, in turn, promotes the interaction of GRK2 with activated receptors and the subsequent receptor phosphorylation, leading to downstream signal transduction, such as ERK signaling and GRK2 degradation (Figure 7). 

Studies have reported that the conformation of GPCRs stabilized by a G-protein-biased agonist differs from that stabilized by a β-arrestin-biased agonist. Notably, the phosphorylation of GPCRs by specific GRKs is necessary for β-arrestin binding. The detailed mapping of β2 adrenergic receptor (β_2_AR) phosphorylation revealed that the β-arrestin-biased ligand (carvedilol) induced a distinct phosphorylation pattern from that of the unbiased full agonist isoproterenol [48]. Biased ligands can recruit different GRKs and induce unique phosphorylation patterns by eliciting distinct conformations of GPCRs. According to the present study, in the conformation that a β-arrestin-biased ligand stabilizes, D_2_R recruits GRK2 in a ubiquitination-dependent manner to mediate receptor phosphorylation for downstream β-arrestin-dependent signal transduction. 

The classic signaling pathway of GPCRs involves agonist receptor binding, which triggers the interaction of the activated receptor with heterotrimeric G proteins. These proteins dissociate into G protein Gα subunits and βγ subunits dimers to facilitate signal transduction. The liberated Gβγ subunits and plasma membrane phospholipids recruit cytoplasmic inactive GRK2 to the vicinity of ligand-bound GPCRs, where they activate GRK2 for receptor phosphorylation. This is followed by the binding of β-arrestins with high affinity to the phosphorylated receptor; however, some functional and structural data have challenged these principles by using altered D2R that are prone to preferentially stimulate G proteins (^[Gprot]^D2R) or recruit β-arrestins (^[βarr]^D2R), in combination with biased ligands that direct wild-type D2R towards β-arrestin in a GRK2 dosage-dependent manner [49]. This study has demonstrated that G-protein-independent GRK2 recruitment is a key factor in preferential D2R-biased signaling towards β-arrestin.

Two different modes of GRK2 ubiquitination have been reported. A more extensively characterized case is Mdm2-mediated ubiquitination, which is induced through the stimulation of β_2_AR and results in the proteasomal degradation of GRK2. It was reported more recently that GRK2 is constitutively ubiquitinated by the Gβ2/DDB1-CUL4A complex and that the agonist stimulation of β_2_AR stabilizes GRK2 protein by dissociating G2 from DDB1-CUL4A [50]. It is not clear at this moment how these conflicting results could be explained. Additionally, previous research has demonstrated that the Mdm2-mediated ubiquitination of p53 and β-arrestin2 occurs in the nucleus [51,52]. According to our results, however, Mdm2-mediated GRK2 ubiquitination occurs in the cytoplasm, even though the nuclear localization of GRK2 has been observed in previous studies. It is worth noting how Mdm2 moves out of the nucleus in response to agonist stimulation; in fact, the subcellular localization of a protein is determined by a dynamic balance between its nuclear import and export pathways [53]. Nuclear import and export are mediated by specific transport proteins that recognize and bind to NLSs or NESs on the protein of interest. The nuclear import pathway is mediated by a family of proteins called importins, which recognize NLSs on the cargo protein and transport it through the nuclear pore complex (NPC) into the nucleus. Once the cargo protein enters the nucleus, it is released from the importin and can interact with other nuclear components [54]. Conversely, the nuclear export pathway is mediated by a family of proteins called exportins, which recognize NESs on the cargo protein and transport it through the NPC to the cytoplasm. The export process is facilitated by the binding of the exportin to the cargo protein, followed by the formation of a trimeric complex with the small GTPase Ran. The trimeric complex then interacts with the NPC and is translocated through the nuclear pore into the cytoplasm. Once in the cytoplasm, the cargo protein is released from the exportin and can interact with other cellular components [55]. The balance between nuclear import and export pathways is regulated by a variety of mechanisms, including the availability of transport proteins, the presence of NLSs and NESs on the protein of interest, and post-translational modifications. Alterations to these mechanisms can lead to changes in the subcellular localization of the protein, which can have profound effects on its cellular function. For example, phosphorylation on a shuttling protein regulates nuclear import or export in either a positive or negative manner [56,57,58]. It is possible that phosphorylation or other posttranslational modifications promote interaction with exportin, which then drives Mdm2 to translocate out of the nucleus. Further studies are needed to understand the precise mechanisms underlying Mdm2-mediated GRK2 ubiquitination and its subcellular localization. 

Ubiquitination encompasses three types of modifications: monoubiquitination, polyubiquitination, and oligoubiquitination. Monoubiquitination refers to the addition of a signal ubiquitin molecule to the lysine residues of a substrate protein. This process regulates nondegradative cellular processes such as membrane trafficking and endocytosis [59]. Additionally, it is implicated in the development of various genetic disorders, including DNA damage [60], mitophagy [61], endosomal sorting [62], and protein localization as well as stability [63]. A substrate protein undergoes polyubiquitination when ubiquitin molecules bind consecutively to a single lysine residue. Among the reported polyubiquitin chains, lysine 48 (K48)-linked chains typically target a protein for proteasomal degradation [64]. In contrast, polyubiquitin chains linked via other lysines, such as K63-linked polyubiquitination, have non-proteolytic functions, such as cellular trafficking [65], DNA damage response [66], and NF-κB activation [67]. Oligoubiquitination specifically refers to the addition of multiple ubiquitin molecules to the same substrate protein, typically in the form of a chain. This process primarily mediates protein stabilization as well as activation, as seen with the example of p53 [68]. It is clear that GRK2 is monoubiquitinated under steady-state conditions and polyubiquitinated during degradation [69]. The present study determined that GRK2 is polyubiquitinated after the activation of GPCRs. It has been shown that the K63-linked ubiquitination of AKT is required for its signaling activity through promoting kinase translocation to the plasma membrane [70]. Thus, it is reasonable to suggest that the Mdm2-mediated ubiquitination of GRK2 could modulate other functional aspects of the kinase, including protein trafficking, subcellular localization, and protein–protein interactions.

Mdm2-mediated ubiquitination promotes the proteasomal degradation of GRK2 [28]. The present study additionally identifies Mdm2-mediated GRK2 ubiquitination as being involved in ERK activation, which is specifically mediated by the D2R β-arrestin signaling pathway. Additionally, a previous study has shown that the phosphorylation of GRK2 on Ser670 via ERKs is necessary for the association of Mdm2 with GRK2 [31]. Our study suggests that Mdm2-mediated GRK2 ubiquitination enhances the activity of GRK2, resulting in the increased phosphorylation of D2R and subsequent activation of downstream signaling pathways, including ERKs. These findings suggest that Mdm2-mediated GRK2 ubiquitination may facilitate an effective positive feedback loop for GRK2-mediated ERK signaling.

In this study, we uncovered the molecular mechanisms implicated in D2R β-arrestin-biased signal-pathway-mediated ERK activation through the regulation of GRK2 ubiquitination. Our results showed that, in response to a D2R β-arrestin-biased activation signal, the ubiquitination of GRK2 occurs in the cytoplasm in an Src-dependent manner, after which the ubiquitinated GRK2 translocates towards the plasma membrane to phosphorylate D2R and mediate ERK signaling. These findings are largely novel and will provide crucial information for comprehending the intricate molecular mechanisms of D2R regulation.

## 4. Materials and Methods

### 4.1. Reagents

DA, MLS1547, rabbit anti-HA antibodies (H6908), rabbit antibodies against green fluorescent protein (GFP) (G1544), rabbit anti-FLAG M2 antibodies (F7425), and agarose beads coated with monoclonal antibodies against FLAG epitope were purchased from Sigma-Aldrich Chemical Co. (St. Louis, MO, USA). PP2 (P125361), while LMB (L102387) were purchased from Aladdin Chemical Reagent Co., Ltd. (Shanghai, China). UNC9994 (cat no. 2562) was purchased form Axon Medchem (Reston, VA, USA). Rabbit antibodies to ERK2 (sc-1647), mouse antibodies against phospho-ERK1/2 (sc7383), and rabbit antibodies against Mdm2 (sc-965) were acquired from Santa Cruz biotechnology, Inc. (Dallas, TX, USA). Antibodies against GRK2 (3982S) were acquired from Cell Signaling Technology (Danvers, MA, USA). Rabbit antibodies to actin (66009-1-LG) were acquired from Proteintech Group, Inc. (Wuhan, China). Anti-rabbit horseradish peroxidase (HRP)-conjugated secondary antibodies (#65-6120) were acquired from Thermo Fisher Scientific (Waltham, MA, USA), and anti-mouse HRP-conjugated secondary antibodies (115-035-003) were acquired from Jackson Immuno Research Laboratories, Inc. (West Grove, PA, USA).

### 4.2. Cell Culture

Human embryonic kidney 293 (HEK 293) cells were acquired from the National Collection of Authenticated Cell Cultures (Shanghai, China) and were cultured (37 °C, 5% CO_2_) in Dulbecco’s modified eagle medium containing 10% fetal bovine serum, 100 units/mL penicillin, and 100 g/mL streptomycin (Thermo Fisher Scientific). Transfections were performed through the use of polyethylenimine (Polyscience, Warrington, PA, USA). The Mdm2-KD and GRK2-KD mutations were generated via stably expressing small hairpin RNAs (shRNAs) in PLKO.1 targeting each puromycin-selected gene. As a negative control, the HEK 293 cell line stably expressing scrambled shRNA in PLK0.1 under puromycin selection was used.

### 4.3. Plasmid Constructs

The mammalian plasmids encoding wild-type human D2R, D2R-IC2/3, GRK2, HA-Ub, Mdm2, and GFP-β-arrestin2 were generously provided by Professor Kyeong-Man Kim (Chonnam National University, Korea). ^[βarr]^D2R (A135R/M140D) and FLAG-tagged, ^[Gprot]^D2R (L125N/Y133L) and FLAG-tagged, ^[D80A]^D2R, FLAG-D2R, and GRK2 encoding plasmids (GFP, FLAG-tagged), and GRK2-4KR, GRK2-3YF, GRK2-K220R, and Mdm2 (GFP, FLAG-tagged) were generated via polymerase chain reaction (TL988/TianLong; China). GRK2 and Mdm2 shRNAs were obtained from Santa Cruz Biotechnology.

### 4.4. Immunoprecipitation

The cells expressing the respective plasmids were lysed in a lysis buffer at 4 °C for 4 h while rotating. The cell lysates were then incubated with agarose beads coated with FLAG antibodies for 2 to 3 h at 4 °C. The beads were rinsed three times for 5 min with an ice-cold washing buffer. Immunoprecipitates (IPs) and cell lysates were separated on SDS-PAGE gels and transferred to nitrocellulose membranes (Sigma-Aldrich chemical Co.). The membranes were incubated overnight at 4 °C with primary antibodies against target proteins, followed by incubation with HRP-conjugated secondary antibodies. A chemiluminescent substrate (Thermo Fisher Scientific) was used to visualize the target proteins. Gray densitometry was used to quantify immunoblots via the use of ImageJ version 1.53c (National Institutes of Health, Bethesda, MD, USA).

### 4.5. Immunocytochemistry

Cells bearing respective plasmids were cultured on glass coverslips. After 24 h, cells were fixed with 4% paraformaldehyde in phosphate-buffered saline (PBS) for 15 min at 25 °C and permeabilized with 0.1% Triton X-100 in PBS for 1 min at the same temperature. Cells were then blocked with PBS containing 3% fetal bovine serum (FBS) and 1% bovine serum albumin (BSA) for 1 h, followed by incubation at room temperature for 2 h with FLAG antibody (1:1000). After three washes with PBS, cells were incubated with Alexa-594-conjugated secondary antibodies (1:500) and observed using a laser-scanning confocal microscope (TCS SP5/AOBS/Tandem; Germany). Each sample was analyzed for approximately 5–10 cells, and the same experiments were repeated three times. ImageJ version 1.53c was used to analyze the images.

### 4.6. ERK1/2 Phosphorylation

After 24–36 h of transfection, a serum-free culture medium was used to deprive the cells overnight. The cells were then treated with 10 μM of DA for 2 min or 10 min, the medium was removed, and the SDS sample buffer was added directly to the culture wells. Following a 20-min incubation at 65 °C, samples were sonicated to shear genomic DNA. Next, the proteins were separated on SDS–PAGE gel and transferred onto nitrocellulose membranes, which were incubated for 1 h at room temperature in TBS-T (50 mM Tris, pH 7.5, 150 mM NaCl, and 0.5% Tween 20) containing 5% non-fat dry milk, followed by overnight incubation with antibodies for p-ERKs or ERK2 (1:1000 dilution) and 1 h with HRP-conjugated secondary antibodies (1:20,000) in 5% non-fat dry milk. A chemiluminescent substrate was used to detect the target proteins. ImageJ version 1.53c (National Institutes of Health, USA) was used to quantify the immunoblots’ gray densitometry.

### 4.7. Reporter Gene Assay

A reporter gene system with a plasmid encoding a firefly luciferase gene controlled via several cAMP responsive elements (CRE; Promega, Madison, WI, USA) as well as a pRL-TK control vector was used to detect the levels of cellular cAMP in an indirect manner. Transfected cells were seeded in 24-well plates with reporter genes, along with ^[WT]^D2R, ^[Gprot]^D2R, ^[βarr]^D2R, or ^[D80A]^D2R plasmids, and each transfection set was organized into three identical groups. ^[WT]^D2R and ^[Gprot]^D2R activation leads to a decrease in intracellular cAMP levels as they couple with Gi/o proteins and mediate AC inhibition. To detect changes in cellular cAMP levels, cells were first exposed to 2 μM forskolin to stimulate cAMP synthesis, after which they were then exposed to the D2R agonist Quin for 4 h at 37 °C, as has been described in previous studies [71,72]. Consistent with the inhibitory role of ^[WT]^D2R and ^[Gprot]^D2R on AC activity, treatment with Quin was able to reduce forskolin-induced cAMP levels in cells. The relative expression of luciferase was determined via the use of a dual luciferase assay reagent (Promega, Madison, WI, USA). The data were presented as a percentage of the cAMP levels stimulated by forskolin. GraphPad Prism version 5.0 (GraphPad software, San Diego, CA, USA) was utilized to fit dose–response curves.

### 4.8. Subcellular Fractionation

According to prior research, cell lysates were divided into cytoplasmic, membrane, and nuclear fractions [73]. Briefly, cells were incubated with buffer-1 (1.5 mM MgCl_2_, 10 mM HEPES/KOH pH 7.8, 0.2 mM phenylmethylsulfonyl fluoride, 0.5 mM dithiothreitol, 10 mM KCl, and 1 mM Na_3_VO_4_) for 20 min, followed by centrifugation at 2000× *g* for 5 min. After an additional 10 min of centrifugation at 15,000× *g*, supernatants were saved as cytoplasmic fractions. Particles from the initial centrifugation were washed with buffer-1 for 15 min and then centrifuged at 15,000× *g* for 10 min. The particles were then centrifuged after being incubated with buffer-2 (0.2 mM EDTA, 20 mM HEPES/KOH pH 7.8, 1.5 mM MgCl_2_, 420 mM NaCl, 0.5 mM dithiothreitol, 25% glycerol, 0.2 mM phenylme-thylsulfonyl fluoride, and 1 mM Na_3_VO_4_) for 20 min. After 10 min of centrifugation at 24,000× *g*, the supernatant was collected as the nuclear extract. β-actin and laminB1 were the reference proteins for the cytosolic and nuclear fractions, respectively.

### 4.9. Statistics

Data from at least three separate experiments were expressed as the mean ± standard deviation (SD). The increase in the gel density value was determined by comparing each value to that of the control group (typically the first lane of the gel). When analyzing the statistical significance of differences between groups, a paired two-tailed Student’s *t*-test and one-way ANOVA with a Tukey post hoc test were used, respectively. A *p*-value less than 0.05 was regarded as significant.

## Figures and Tables

**Figure 1 ijms-24-10031-f001:**
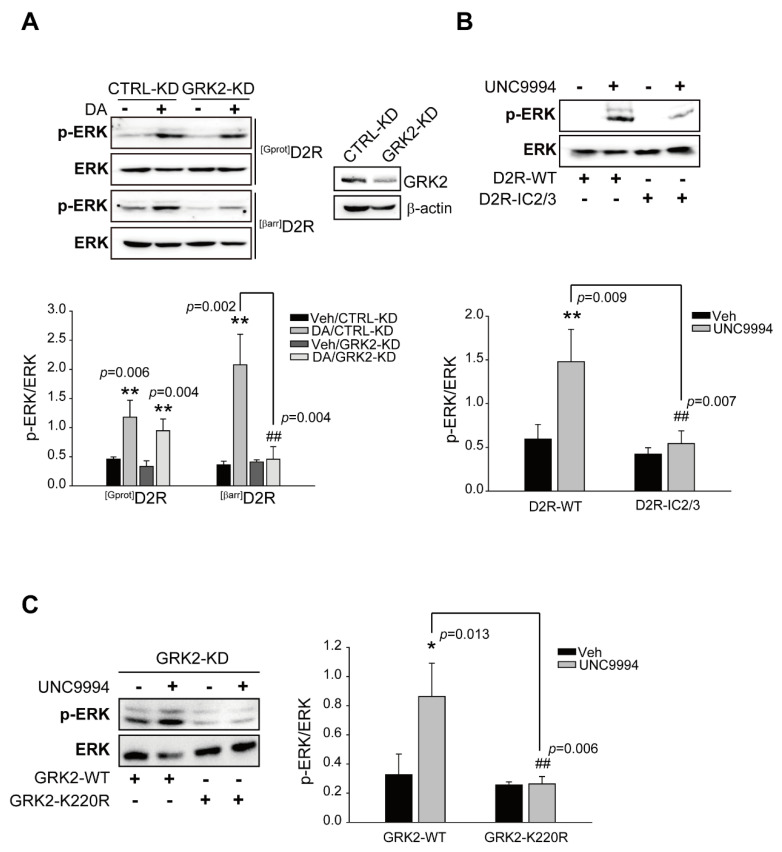
Functional roles of GRK2-mediated receptor phosphorylation in D2R G protein or D2R β-arrestin pathway-mediated ERK activation. (**A**) CTRL-KD and GRK2-KD cells were transfected with plasmids encoding ^[Gprot]^D2R or ^[βarr]^D2R. Serum-starved cells were stimulated with 10 μM DA for 2 min (^[Gprot]^D2R-producing) or 10 min (^[βarr]^D2R-producing). Cell lysates from CTRL-KD and GRK2-KD cells were immunoblotted with antibodies against p-ERK1/2 (1:1000 dilution) and ERK2 (1:1000 dilution), respectively. Cell lysates were immunoblotted with GRK2 or β-actin antibodies. Each KD cell contained approximately 75% less cellular GRK2. We measured the levels of phosphorylated ERKs (p-ERKs) and total ERKs in the same sample and then divided the amount of p-ERKs by the amount of total ERKs to obtain the p-ERK/ERK ratio. The p-ERK/ERK ratio provided a normalized measure of ERK pathway activation. ** *p* < 0.01 compared with the corresponding Veh group, ^##^ *p* < 0.01 compared with the DA/CTRL-KD group (*n* = 3). (**B**) HEK 293 cells were transfected with plasmids encoding D2R-WT or D2R-IC2/3. Serum-starved cells were incubated in the presence of 1 μM UNC9994 for 10 min. Lysates were immunoblotted with anti-p-ERK1/2 (1:1000 dilution) and anti-ERK2 (1:1000 dilution) antibodies, respectively. ** *p* < 0.01 compared with the Veh group, ^##^ *p* < 0.01 compared with the D2R/UNC9994 group (*n* = 3). (**C**) GRK2-KD cells producing D2R were transfected with plasmids encoding GRK2-WT or GRK2-K220R. Serum-starved cells were incubated in the presence of 1 μM UNC9994 for 10 min. Lysates were immunoblotted with anti-p-ERK1/2 (1:1000 dilution) and anti-ERK2 (1:1000 dilution) antibodies, respectively. * *p* < 0.05 compared with the Veh group, ^##^ *p* < 0.01 compared with the WT/UNC9994 group (*n* = 3).

**Figure 2 ijms-24-10031-f002:**
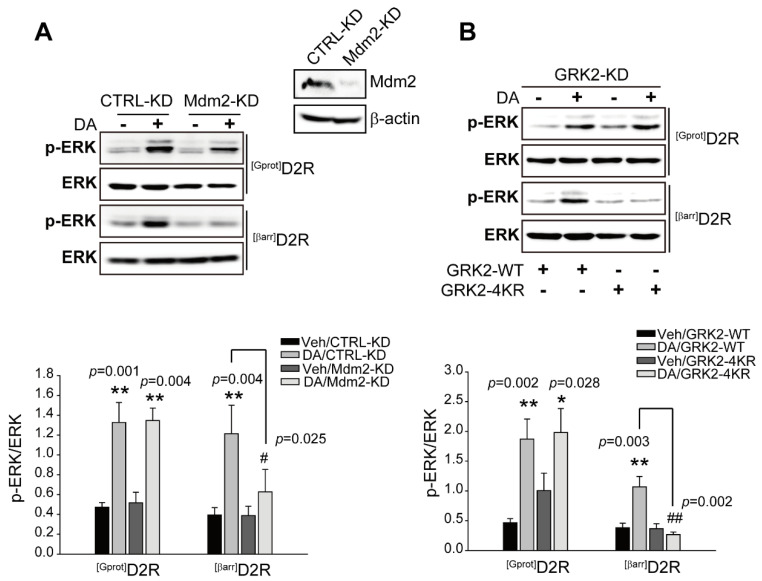
The ubiquitination of GRK2 is involved in D2R β-arrestin pathway-mediated ERK activation. (**A**) CTRL-KD and Mdm2-KD cells were transfected with plasmids encoding ^[Gprot]^D2R or ^[βarr]^D2R. Serum-starved cells were stimulated with 10 μM DA for 2 min (^[Gprot]^D2R producing) or 10 min (^[βarr]^D2R producing). CTRL-KD and Mdm2-KD cell lysates were immunoblotted using p-ERK1/2 (1:1000 dilution) and ERK2 (1:1000 dilution) antibodies, respectively. Cell lysates were immunoblotted with Mdm2 or β-actin antibodies. About 86% of the Mdm2 levels in cells were diminished. ** *p* < 0.01 compared with the corresponding Veh group, ^#^ *p* < 0.05 compared with the DA stimulation group (*n* = 3). (**B**) GRK2-KD cells were transfected with plasmids encoding ^[Gprot]^D2R or ^[βarr]^D2R, and co-transfected with plasmids encoding GRK2-WT or GRK2-4KR. Serum-starved cells were treated with 10 μM DA for 2 min (^[Gprot]^D2R-producing) or 10 min (^[βarr]^D2R-producing). Cells lysates were immunoblotted using p-ERK1/2 (1:1000 dilution) and ERK2 (1:1000 dilution) antibodies, respectively. We measured the levels of p-ERKs and total ERKs in the same sample and then divided the amount of pERKs by the amount of total ERKs to obtain the p-ERK/ERK ratio. The pERK/ERK ratio provided a normalized measure of ERK pathway activation. ** *p* < 0.01, * *p* < 0.05 compared with the corresponding Veh group, ^##^ *p* < 0.01 compared with the DA/GRK2-WT/^[βarr]^D2R expression group (*n* = 3).

**Figure 3 ijms-24-10031-f003:**
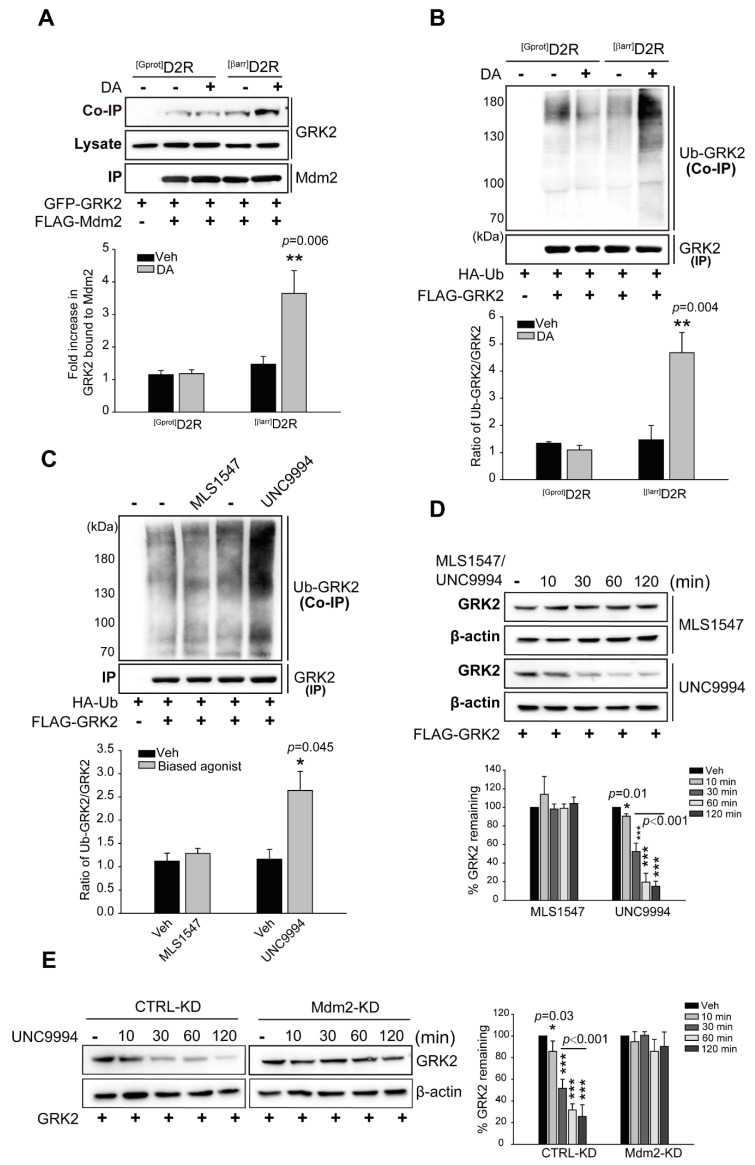
The activation of the D2R β-arrestin pathway promotes GRK2 ubiquitination. (**A**) HEK 293 cells producing ^[Gprot]^D2R or ^[βarr]^D2R were transfected with plasmids encoding GFP-GRK2 and FLAG-Mdm2. The cells were stimulated with either Veh or 10 μM DA for 2 min. FLAG beads were used to immunoprecipitate cell lysates. Co-IP/Lysate and IP were immunoblotted via the use of GFP (1:1000 dilution) and FLAG (1:1000 dilution) antibodies, respectively. The data represent the outcomes of three independent investigations with comparable results. ** *p* < 0.01 compared with the Veh group (*n* = 3). (**B**) HEK 293 cells were transfected with plasmids encoding ^[Gprot]^D2R or ^[βarr]^D2R, HA-Ub, and FLAG-GRK2. The cells were treated for 2 min with either a vehicle or 10 μM DA. Immunoprecipitation of cell lysates using FLAG beads. Antibodies against HA (1:1000 dilution) and FLAG (1:1000 dilution) were used to immunoblot Co-IP and IP. ** *p* < 0.01 compared with the Veh group (*n* = 3). (**C**) HEK 293 cells producing D2R were transfected with plasmids encoding HA-Ub and FLAG-GRK2. The cells were stimulated with either 10 μM MLS1547 or 1 μM UNC9994 for 2 min. Immunoprecipitation of cell lysates using FLAG beads. Antibodies against HA (1:1000 dilution) and FLAG (1:1000 dilution) were used to immunoblot Co-IP and IP. * *p* < 0.05 compared with the Veh group (*n* = 3). (**D**) HEK 293 cells were transfected with plasmids encoding D2R and FLAG-GRK2. Cells were prestimulated with 50 μg/mL cycloheximide for 1 h, followed by 10 μM MLS1547 or 1 μM UNC9994 treatment for 0–2 h. Cell lysates were immunoblotted using FLAG (1:1000 dilution) or β-actin (1:2000 dilution) antibodies. “0 min” groups were normalized to 100%. * *p* < 0.05, *** *p* < 0.001 compared with the Veh group (*n* = 3). (**E**) CTRL-KD and Mdm2-KD cells producing D2R were transfected with plasmids encoding GRK2. The pretreatment of cells with 50 μg/mL cycloheximide for 1 h was followed with treatment with 1 μM UNC9994 for 0–2 h. The immunoblotting of cell lysates with antibodies against GRK2 (1:2000 dilution) or β-actin (1:2000 dilution) was performed. “0 min” groups were normalized to 100%. * *p* < 0.05, *** *p* < 0.001 compared with the Veh/CTRL-KD group (*n* = 3).

**Figure 4 ijms-24-10031-f004:**
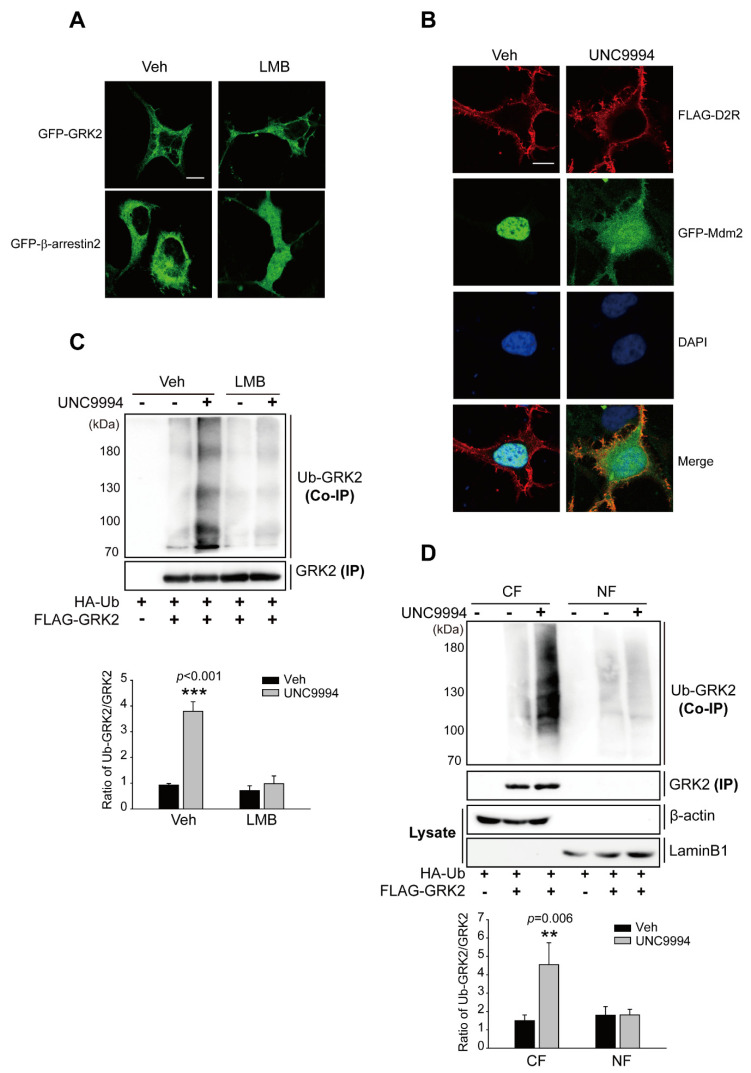
The Mdm2-mediated ubiquitination of GRK2 occurs in the cytoplasm in response to UNC9994 stimulation. (**A**) HEK 293 cells were transfected with plasmids encoding GFP-GRK2 or GFP-β-arrestin2. Cells were exposed to a vehicle or 10 ng/mL LMB for 3 h. The horizontal bar represents 10 μm. One representative example from three independent investigations is depicted in the data. (**B**) HEK 293 cells were transfected with plasmids encoding FLAG-D2R and GFP-Mdm2. Cells were stimulated with either a vehicle or 1 μM UNC9994 for 10 min. Later, the cells were incubated with FLAG antibodies (1:1000 dilution) and Alexa-594-conjugated anti-rabbit secondary antibodies (1:500 dilution) in succession. The horizontal bar represents 10 μm. One representative example from three independent investigations is depicted in the data. (**C**) HEK 293 cells were transfected with plasmids encoding D2R, HA-Ub, and FLAG-GRK2. Cells were pretreated with 10 ng/mL LMB for 3 h, followed by 1 μM UNC9994 treatment for 2 min. Cell lysates were immunoprecipitated via the use of FLAG beads. Co-IP and IP were immunoblotted with HA (1:1000 dilution) and FLAG (1:1000 dilution) antibodies, respectively. *** *p* < 0.001 compared with the Veh/veh group (*n* = 3). (**D**) HEK 293 cells were transfected with plasmids encoding D2R. Cells were stimulated with 1 μM UNC9994 for 2 min. The fractionation of cell lysates followed the protocol outlined in the “Section 4”. Nuclear and cytosolic fractions were utilized in ubiquitination assays. NF, nuclear fraction; CF, cytosolic fraction. ** *p* < 0.01 compared with the Veh/CF group (*n* = 3).

**Figure 5 ijms-24-10031-f005:**
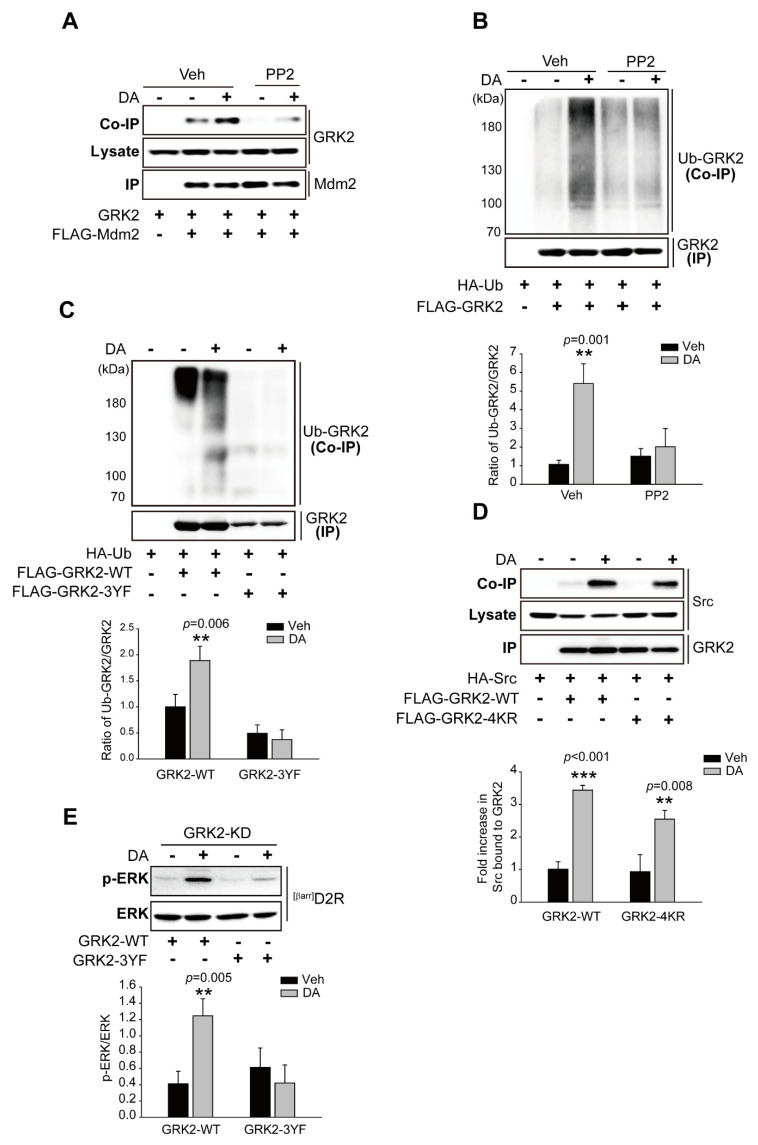
The tyrosine phosphorylation of GRK2 is required for Mdm2-mediated GKR2 ubiquitination upon the stimulation of the D2R β-arrestin-dependent pathway. (**A**) HEK 293 cells were transfected with plasmids encoding ^[βarr]^D2R, GRK2, and FLAG-Mdm2. The pretreatment of cells with 10 μM PP2 for 30 min was followed by a 2-min treatment with 10 μM DA. The immunoprecipitation of cell lysates using FLAG beads. Co-IP/Lysate and IP were immunoblotted, respectively, with antibodies against GRK2 (1:2000) and FLAG (1:1000). (**B**) HEK 293 cells were transfected with plasmids encoding ^[βarr]^D2R, HA-Ub, and FLAG-GRK2. Cells were pretreated with 10 μM PP2 for 30 min, followed by 10 μM DA treatment for 2 min. The immunoprecipitation of cell lysates using FLAG beads. Co-IP/Lysate and IP were immunoblotted, respectively, with antibodies against HA (1:1000 dilution) and FLAG (1:1000 dilution). ** *p* < 0.01 compared with the Veh group (*n* = 3). (**C**) HEK 293 cells were transfected with plasmids encoding ^[βarr]^D2R, HA-Ub, and FLAG-GRK2-WT or FLAG-GRK2-3YF. The cells were treated with either a vehicle or 10 μM DA for 2 min. Immunoprecipitation of cell lysates using FLAG beads. Co-IP/Lysate and IP were immunoblotted with antibodies against HA (1:1000 dilution) and FLAG (1:1000 dilution), respectively. ** *p* < 0.01 compared with the Veh group (*n* = 3). (**D**) HEK 293 cells were transfected with plasmids encoding ^[βarr]^D2R, HA-Src, and FLAG-GRK2-WT or FLAG-GRK2-4KR. Cells were treated with either a vehicle or 10 μM DA for 2 min. The immunoprecipitation of cell lysates using FLAG beads. HA (1:1000 dilution) and FLAG (1:1000 dilution) antibodies were used to immunoblot Co-IP/Lysate and IP, respectively. ** *p* < 0.01, *** *p* < 0.001 compared with corresponding Veh group (n = 3). (**E**) GRK2-KD cells producing ^[βarr]^D_2_R were transfected with plasmids encoding GRK2-WT or GRK2-3YF. Serum-starved cells were incubated in the presence of 10 μM DA for 10 min. Antibodies against p-ERK1/2 (1:1000 dilution) and ERK2 (1:1000 dilution) were used to immunoblot lysates. ** *p* < 0.01 compared with the corresponding Veh/WT group (*n* = 3).

**Figure 6 ijms-24-10031-f006:**
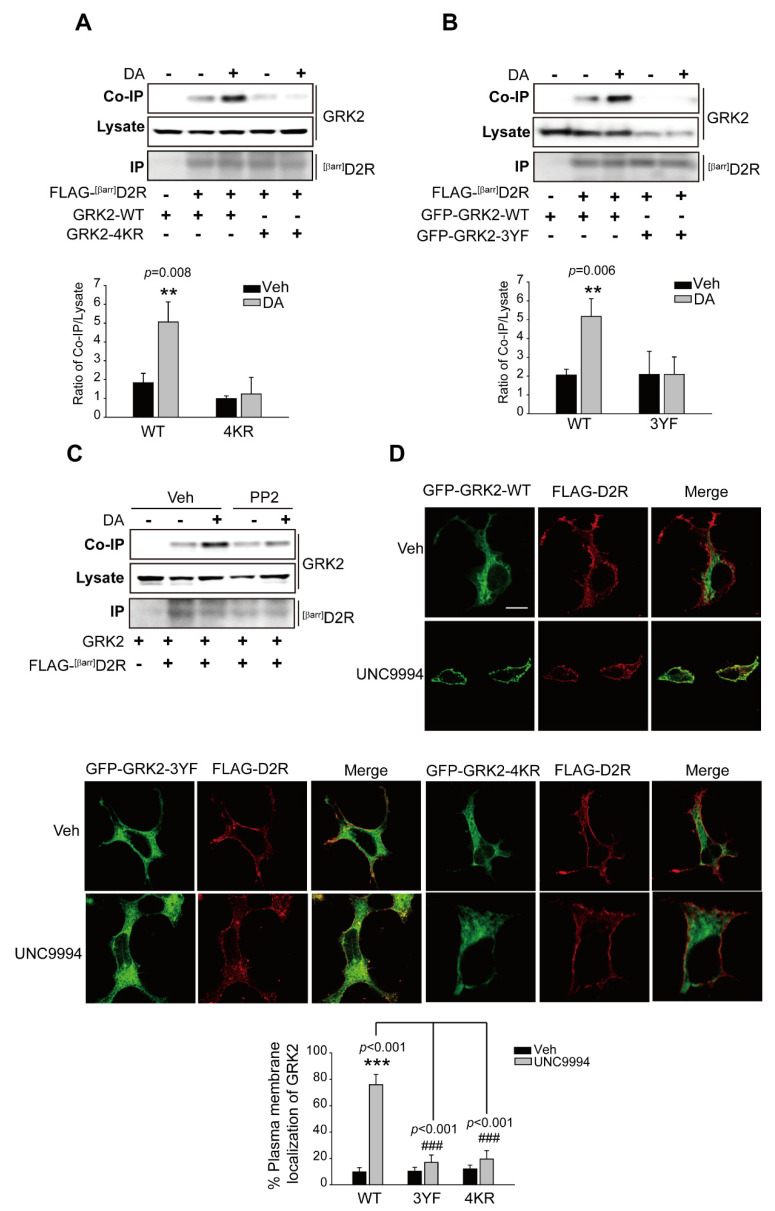
The translocation of GRK2 to the plasma membrane and subsequent interaction with activated D2R depend on its ubiquitination. (**A**) HEK 293 cells were transfected with plasmids encoding FLAG-^[βarr]^D2R, and co-expressed with GRK2-WT or GRK2-4KR. The cells were treated for 2 min with either a vehicle or 10 μM DA. Immunoprecipitation of cell lysates using FLAG beads. Antibodies against GRK2 (1:2000 dilution) and FLAG (1:1000 dilution) were used to immunoblot Co-IP/Lysate and IP, respectively. ** *p* < 0.01 compared with the corresponding Veh/WT group (*n* = 3). (**B**) HEK 293 cells producing FLAG-^[βarr]^D2R were transfected with plasmids encoding GFP-GRK2-WT or GFP-GRK2-3YF. The cells were treated for 2 min with either a vehicle or 10 μM DA. The immunoprecipitation of cell lysates using FLAG beads. Antibodies against GFP (1:1000 dilution) and FLAG (1:1000 dilution) were used to immunoblot Co-IP/Lysate and IP, respectively. ** *p* < 0.01 compared with the corresponding Veh/WT group (*n* = 3) (**C**) HEK 293 cells were transfected with plasmids encoding ^[βarr]^D2R and GRK2. Cells were pretreated with 10 μM PP2 for 30 min, followed by 10 μM DA treatment for 2 min. Immunoprecipitation of cell lysates using FLAG beads. Antibodies against GRK2 (1:2000 dilution) and FLAG (1:1000 dilution) were used to immunoblot Co-IP/Lysate and IP, respectively. (**D**) HEK 293 cells were transfected with plasmids encoding FLAG-D2R and GFP-GRK2-WT, GFP-GRK2-4KR, or GFP-GRK2-3YF. The cells were treated for 2 min with either a vehicle or 1 μM UNC9994. The cells were incubated with FLAG antibodies (1:1000 dilution) and Alexa-594-conjugated anti-rabbit secondary antibodies (1:500 dilution) subsequently. Based on four–five images for each panel, the ratios of fluorescence signals detected in the entire cell region (cytoplasm plus plasma membrane) and plasma membrane were calculated before and after UNC9994 treatment to quantify the extent of GRK2 translocation into the plasma membrane as a result of UNC9994 treatment. *** *p* < 0.001 compared to the Veh/WT group, ^###^
*p* < 0.001 compared to the WT/UNC9994 group. Magnification 100×; horizontal bar represents 10 μm. One representative example from three independent investigations is depicted in the data.

**Figure 7 ijms-24-10031-f007:**
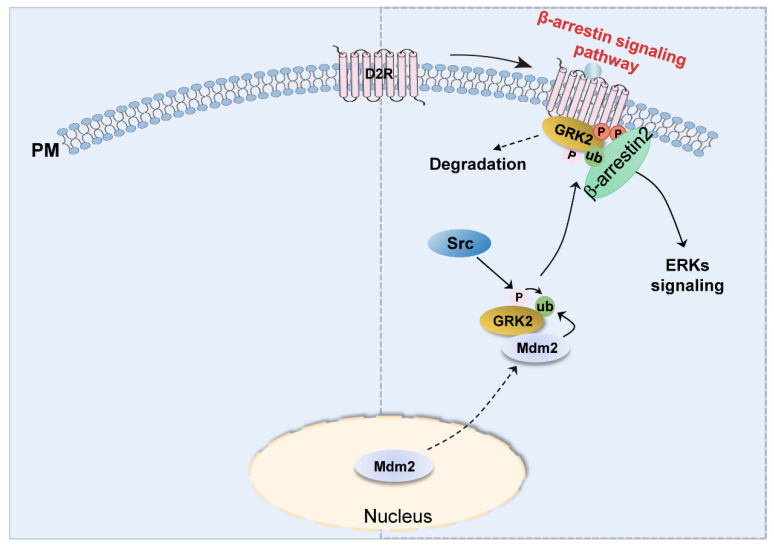
Diagram showing the mechanisms involved in D2R β-arrestin-dependent pathway-mediated ERK activation. After stimulation with an agonist to activate the D2R β-arrestin signaling pathway, Mdm2 moves out of the nucleus to ubiquitinate GRK2, which is in an Src-dependent tyrosine phosphorylation state. Ubiquitinated GRK2 then translocates to the plasma membrane and interacts with activated D2R, followed by the phosphorylation of D2R and recruiting β-arrestin to mediate downstream ERK signal transduction.

## Data Availability

The raw data presented in this study are available on request from the corresponding author.

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
