# Peer review of "Ubiquitination of GRK2 Is Required for the β-Arrestin-Biased Signaling Pathway of Dopamine D2 Receptors to Activate ERK Kinases"

_ijms, 2023, doi:10.3390/ijms241210031_

Round 1
Reviewer 1 Report
This research article describes the molecular mechanisms involved in the D2R β-arrestin-biased signal pathway-mediated ERKs activation through regulation of GRK2 ubiquitination. These findings provide essential information for understanding the detailed molecular mechanisms of D2R regulation and signaling.
- (Fig 1) It seems unusual to use the same antibody dilution (1:1000) for detecting both pERKs and ERKs, as ERKs are typically more abundant and antibody may work well at a more diluted concentration. Why choose an specific antibody for total ERK2 instead of total ERK1/2?
- Individual values of quantification should be included on the graph bars.
- One of the main limitations of the study is that all the assays were performed in transfected cells. It may be worth considering testing this mechanism of interaction between GRK2 and D2R is linked to ß-arrestins signaling pathway in an experimental model that is closer to physiological conditions, such as a primary culture, to further validate the findings of the study or to find other proteins implicated in the signaling regulation that maybe are not present in HEK.
- Furthermore, although the study provides valuable information on the molecular mechanisms involved in D2R signaling regulation, additional studies may be needed to confirm and extend these findings.
For example, the D2R receptor can recruit different types of ß-arrestin (DOI: 10.1124/mol.119.115998, https://pubmed.ncbi.nlm.nih.gov/12869650/) it would be interesting to determine which ones favors ERK phophorylation and are related to the GRK2 interaction with D2R.
Reviewer 2 Report
File attached

Nomenclature needs corrctions
Round 2
Reviewer 2 Report
File attached
